# International Delphi consensus on acute kidney injury: Foundations for AI-driven digital twin development in critical care nephrology

Mehdi Kashani[1], Jacob Ninan[2], Lifang Wei[1], Wisit Cheungpasitporn[1], Amos Lal[3], Ognjen Gajic[3], Kianoush Kashani[1,3]*

**1** Division of Nephrology and Hypertension, Department of Medicine, Mayo Clinic, Rochester, Minnesota, United States of America, **2** Division of Critical Care and Nephrology, Department of Medicine, MultiCare Capital Medical Center, Olympia, Washington, United States of America, **3** Division of Pulmonary and Critical Care Medicine, Department of Medicine, Mayo Clinic, Rochester, Minnesota, United States of America

* kashani.kianoush@mayo.edu

## Abstract

### Background

Acute kidney injury (AKI) in critically ill patients is clinically complex and heterogeneous, limiting the development of structured simulation models. Digital twin approaches require clearly defined causal relationships grounded in expert consensus. We aimed to establish an international Delphi consensus to inform the development of an interpretable, AI-driven digital twin framework for AKI in critical care.

### Methods

We conducted a modified Delphi study involving up to three survey rounds. Experts in nephrology and critical care were invited to evaluate statements addressing AKI etiology, biomarker integration, biopsy indications, contrast use, and ICU management. Consensus was predefined as ≥75% agreement.

### Results

Fifty-six experts were invited, and 49 completed the first round. Consensus was achieved after two rounds. Hemodynamic instability and nephrotoxicity were identified as leading contributors to AKI. Experts supported integrating biomarkers beyond serum creatinine and urine output, although agreement varied for specific assays. Daily biomarker assessment reached consensus, whereas contrast use in advanced AKI stages did not. Desmopressin use prior to kidney biopsy in patients with markedly elevated blood urea nitrogen achieved consensus. A structured Directed Acyclic Graph was developed to represent the expert-derived causal framework.

**Data availability statement:** All relevant data are within the manuscript and its Supporting information files.

**Funding:** The author(s) received no specific funding for this work.

**Competing interests:** The authors have declared that no competing interests exist.

## Conclusions

This Delphi study established a clinically grounded framework for AKI in critical care while highlighting areas of practice variability. The resulting causal structure provides a transparent foundation for future AI-driven digital twin development and prospective validation.

## Introduction

Acute Kidney Injury (AKI) remains a major challenge in critical care medicine, contributing to considerable morbidity, mortality, and long-term health burdens among hospitalized patients. Its prevalence in intensive care units highlights the urgency for timely recognition and effective management strategies. Despite the widespread implementation of clinical guidelines such as those from the Kidney Disease: Improving Global Outcomes (KDIGO) initiative, substantial variability persists in the diagnosis and treatment of AKI, resulting in inconsistent care and adverse patient outcomes [1–3].

This inconsistency underscores the need for a more unified, predictive, and clinically informed framework. In recent years, the concept of digital twins—virtual representations of real-world patient physiology that are continuously updated by clinical data—has emerged as a promising approach in precision medicine. These models enable simulation of disease progression, guide therapeutic interventions, and support real-time decision-making, thereby reducing patient risk by prospectively modeling therapeutic scenarios without exposing patients to unnecessary harm [3–5]. Originally developed for industrial applications, digital twins have begun to demonstrate transformative value in healthcare, particularly in complex conditions that require nuanced, data-driven care strategies, such as AKI [6–8].

In the context of this manuscript, the term "digital twin" refers to emerging clinical digital twin frameworks that integrate expert-defined causal structures with real-world clinical data to support prediction, monitoring, and decision-making guidance. This approach differs from traditional mechanistic physiological simulators, which rely primarily on mathematical modeling of physiological processes. Recent work in critical care has demonstrated the feasibility of hybrid digital twin systems that combine structured clinical knowledge, data-driven components, and dynamic updating through patient-specific information. Our Delphi process aims to provide the clinical reasoning and causal architecture necessary for this form of hybrid digital twin development in AKI.

The development of clinically meaningful digital twins relies on capturing domain expertise in a structured, transparent manner. Expert consensus is critical for identifying the clinical variables, pathophysiological mechanisms, and treatment pathways that should inform predictive modeling. The Delphi method, a structured communication process, facilitates this by engaging a panel of experts in iterative feedback to achieve consensus on key elements of care. This methodology has been successfully applied in high-risk fields such as surgery, critical care, and nephrology to define standards and guide quality improvement initiatives [9].

In addition to consensus-building, integrating Directed Acyclic Graphs (DAGs) into this process provides a clearer visual representation of the causal relationships between risk factors and outcomes in AKI. DAGs help formalize clinical reasoning, making the underlying structure of digital twin algorithms more interpretable, reproducible, and aligned with clinical practice [10–14].

By uniting expert clinical insight with advanced modeling approaches, this initiative aims to establish a validated foundation for digital twin development in AKI. The result is a framework that not only supports individualized care and early intervention but also enhances the transparency and clinical relevance of AI-driven decision support tools.

## Methods

We conducted a modified Delphi study to develop expert consensus on key components of Acute Kidney Injury (AKI), focusing on its pathophysiology, diagnostic considerations, and management strategies. The Delphi technique is a well-established methodology that facilitates structured communication among subject matter experts through multiple rounds of questionnaires and feedback. It is particularly useful in areas where clinical uncertainty remains, and expert consensus is needed to guide practice and innovation [15,16].

### Study design

This study was designed as a three-round modified Delphi process. Consensus was predefined as at least 75% of participants rating a statement 8 or higher on a 10-point Likert scale. The process was structured to continue for up to three rounds, depending on whether the consensus threshold was achieved.

The 75% agreement threshold was selected because it is widely used in Delphi studies within nephrology, critical care, and healthcare quality research. This level provides a balanced, methodologically rigorous definition of agreement, securing substantial expert support while avoiding overly restrictive cutoffs that may limit consensus on clinically meaningful items.

The overall flow of the Delphi process is illustrated in Fig 1.

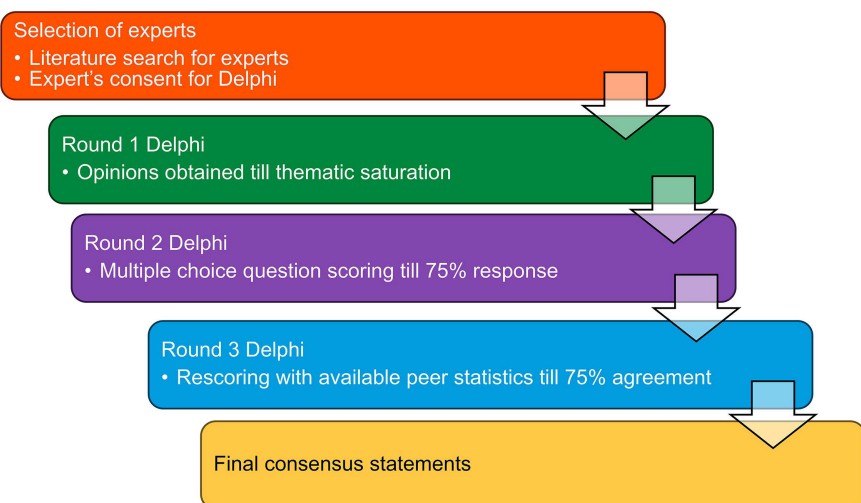

**Fig 1. Flow chart for the Delphi model for consensus generation.**

## Participant selection and recruitment

Experts were selected based on their clinical experience and academic contributions in nephrology and critical care. Candidates were identified through literature review, academic publication databases, and professional networks. Eligible participants had demonstrated subject matter expertise, evidenced by peer-reviewed publications, involvement in guideline development, or long-standing clinical practice in intensive care nephrology. Invitations included detailed information about the study objectives, participation requirements, and confidentiality. Written informed consent was obtained before participation.

A total of 56 experts expressed interest. Of those, 49 individuals completed the first-round survey, yielding an 87.5% response rate. Participants' geographic distribution included representatives from North America, South America, Europe, Asia, and Australia. A full visualization of global participation is provided in Fig 2.

The panel comprised nephrologists, intensivists, and dual-trained clinicians with demonstrated expertise in AKI care and research. Experts in engineering or data science were not included, as this Delphi process was specifically designed to elicit clinically grounded reasoning and domain expertise to inform subsequent model development.

## Delphi procedure

**Round 1** involved a structured questionnaire delivered via REDCap (Research Electronic Data Capture). Statements were presented across major AKI domains, including etiology, diagnostic biomarkers, and ICU management practices. Participants rated each item on a 10-point Likert scale and were encouraged to provide qualitative feedback. The Round 1 questionnaire is included in S1 File.

Based on the responses and comments, statements were revised for clarity or excluded if consensus appeared unlikely. In **Round 2**, these refined statements were re-distributed along with group-level feedback, including median scores, inter-quartile ranges, and summary statistics. This approach allowed participants to reassess their positions in the context of collective feedback. The Round 2 instrument is provided in S2 File. S3 File provides a set of de-identified information.

# GEOGRAPHIC DISTRIBUTION OF EXPERT PARTICIPATION

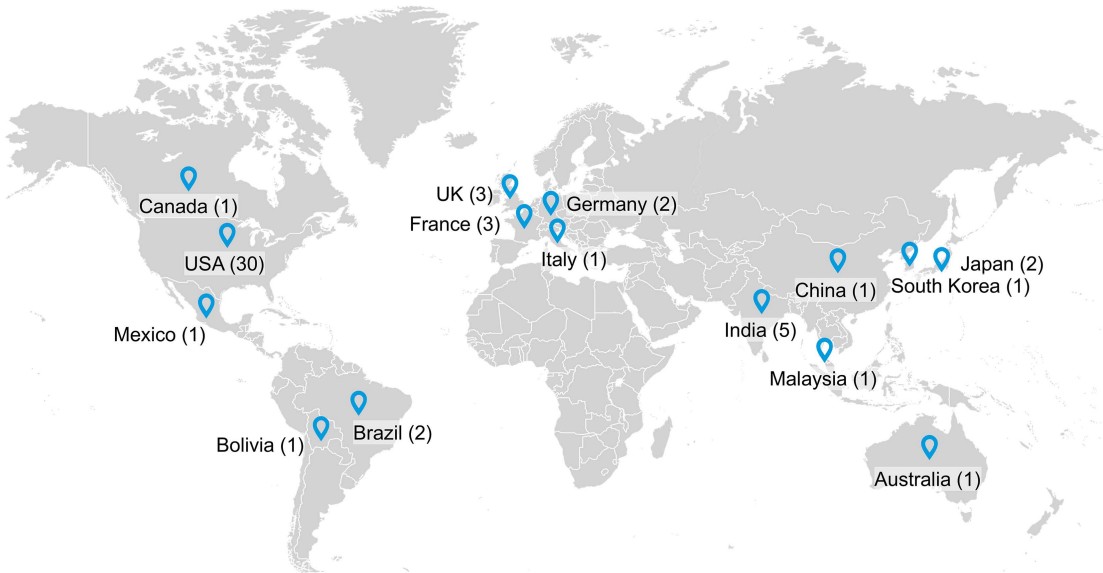

**Fig 2. Geographic Distribution of Expert Participation in the Delphi Study.** This map illustrates the global representation of experts who contributed to the Delphi consensus on Acute Kidney Injury (AKI).

### Data analysis

Quantitative data from both rounds were analyzed using descriptive statistics, including measures of central tendency and variability. Qualitative feedback from Round 1 was subjected to thematic analysis to capture recurring themes, clarify disagreements, and guide revisions.

While the Delphi process formed the backbone of expert consensus, we also aimed to create a framework for clinical modeling. To achieve this, we used simplified Directed Acyclic Graphs (DAGs) to map causal relationships among AKI-related variables. The DAG approach helps translate expert insight into a structured visual model, facilitating interpretability for subsequent digital twin development.

## Results

### Delphi panel composition and participation

A total of 56 experts initially expressed interest in participating, and 49 completed the first-round survey, resulting in an 87.5% response rate. Of the seven who did not participate, five did not complete the survey, and two indicated that the content was outside their scope of expertise. The 49 participating experts represented a diverse mix of specialties: 28 (57%) were nephrologists, 12 (24%) were intensivists, and 16 (32%) had dual specialization in both nephrology and critical care.

In terms of clinical experience, 3 participants (6%) had less than 5 years of practice, 8 (15%) had 5–10 years, 22 (40%) had 10–20 years, and another 22 (40%) had over 20 years of experience. Most experts were affiliated with academic institutions (87%), with the remaining 11% in private practice and 4% from Veterans Affairs hospitals. The panel included a broad geographical representation, with participants from the United States (30), Canada (1), Mexico (1), Brazil (2), Bolivia (1), the United Kingdom (3), France (3), Germany (2), Italy (1), Australia (1), and various countries in Asia, including India (5), Japan (2), China (1), South Korea (1), and Malaysia (1). This global representation is shown in Fig 2.

### First round of Delphi survey

The Round 1 questionnaire, administered via REDCap (see S1 File), asked participants to rate various causes, diagnostic tools, and treatment approaches for AKI in ICU patients. An agreement was defined as a rating of 8 or higher on a 10-point scale, and consensus was considered achieved when at least 75% of experts provided such a rating.

### Common causes of AKI in the ICU

Hemodynamic instability received the highest level of agreement among experts as a leading cause of AKI in the survey, receiving a total score of 68%. Half of the experts ranked it as the top cause, and an additional 26% ranked it second. Inflammation followed with a total score of 50%, with 26% ranking it first and 22% ranking it second. Decreased effective blood volume was ranked third, with a composite score of 37%; although only 7% ranked it as the most important cause, 26% and 17% ranked it second and third, respectively.

Nephrotoxicity was ranked fourth overall, scoring 30%. Forty-four percent of experts ranked it third in importance, and 35% placed it fourth. Other contributors included urinary obstruction (30%), intratubular obstruction (22%), and acute interstitial nephritis (AIN) (21%).

### Biomarker use in AKI management

Most experts strongly supported incorporating biomarkers beyond serum creatinine and urine output into the evaluation of acute kidney injury (AKI) in critically ill patients. A large majority considered access to biomarker testing essential for improving diagnostic accuracy and refining risk stratification.

Among the assays most familiar to participants were neutrophil gelatinase–associated lipocalin (NGAL) and cystatin C, which continue to be widely used where available. When asked to identify markers with the greatest potential

for early AKI detection, experts most frequently highlighted cell cycle arrest biomarkers, specifically tissue inhibitor of metalloproteinases-2 (TIMP-2) and insulin-like growth factor–binding protein 7 (IGFBP7), followed by urinary and plasma NGAL. These markers, along with kidney injury molecule-1 (KIM-1), were also viewed as helpful for prognostic assessment.

At the same time, experts acknowledged that the availability of advanced biomarkers varies substantially across centers. This variability likely explains differences in agreement regarding specific assays, even for those with strong published performance characteristics, such as the combined TIMP-2·IGFBP7 test. The results, therefore, reflect both the promise of biomarker integration and the practical limitations that continue to shape real-world implementation.

## Biomarker testing frequency

There was strong consensus regarding testing frequency: 84% of experts recommended biomarker assessment every 24 hours, and 54% supported testing every 48 hours. Less frequent testing (every 72 hours or weekly) was generally considered inappropriate for high-risk ICU patients.

## Urine indices

Among urine indices, the blood urea nitrogen–to–creatinine ratio was ranked highest in diagnostic value (45%), followed by urine osmolality (35%) and spot urinary sodium (31%). Fractional excretion of sodium (FeNa) and fractional excretion of urea (FeUrea) were less favored.

## Indications for kidney biopsy

The leading indication for kidney biopsy was proteinuria greater than 3 g/day (52%), followed by dysmorphic red blood cells (RBCs) in the urine (45%) and suspected acute interstitial nephritis (AIN) (41%). For progressive acute kidney injury (AKI) without a clear etiology, 48% of experts reported that they rarely performed biopsies. Non-oliguric AKI, isolated creatinine elevation, and unexplained increases in serum creatinine were infrequently considered sufficient indications for biopsy.

Regarding timing, most experts preferred performing a biopsy 7–14 days after AKI onset, whereas early biopsy within 48 hours was rarely endorsed.

## Contrast use in CKD and AKI patients

Experts were most cautious about iodinated contrast use in patients with chronic kidney disease (CKD) stages 4 or 5, where agreement for contrast administration was low. In contrast, there was greater acceptance of contrast use in CKD stages 2 and 3, particularly in the absence of significant proteinuria.

For acute kidney injury (AKI), contrast administration in KDIGO stage 1 was generally considered acceptable when clinically necessary. In KDIGO stage 2, opinions were divided and contrast was typically reserved for essential procedures on a case-by-case basis. In KDIGO stage 3 AKI, agreement for contrast use was low (approximately 30%), reflecting substantial caution among experts in this high-risk population.

Because no AKI stage reached the predefined 75% consensus threshold, contrast use in AKI was not included among the final consensus statements summarized in Table 1.

## Anemia management in the ICU

Blood transfusion was the preferred intervention, followed by epoetin (31%) and darbepoetin (18%). IV iron was used selectively, and newer agents, such as roxadustat or peginesatide, were rarely recommended.

**Table 1.** Summary of expert consensus statements on acute kidney injury in the ICU based on a modified Delphi process.

| Domain | Delphi Question/ Statement | Response Format | Round Evaluated | Agreement (%) | Consensus Achieved (≥75%) | Notes |
|---|---|---|---|---|---|---|
| Etiology of AKI | Hemodynamic instability is a leading cause of AKI in the ICU | Likert (≥8/10) | Round 1 | 76% | Yes | Ranked #1 overall |
| | Nephrotoxicity is a leading cause of AKI | Likert (≥8/10) | Round 2 | 96% | Yes | Achieved consensus after revision |
| | Inflammation is a major contributor to AKI | Likert | Round 1 | 72% | No | High agreement but below threshold |
| | Decreased effective blood volume contributes to AKI | Likert | Round 1 | 67% | No | Did not meet consensus threshold |
| | Urinary obstruction contributes to AKI | Likert | Round 1 | 30% | No | Limited support |
| Biomarker Integration | Incorporation of additional biomarkers beyond creatinine improves AKI evaluation | Likert | Round 1 | 85% | Yes | Strong support for biomarker integration |
| | Daily biomarker testing (every 24 hours) is appropriate in ICU patients | Likert | Round 1 | 84% | Yes | Preferred frequency |
| | Cell cycle arrest markers (TIMP-2·IGFBP7) are valuable in AKI assessment | Ranking | Round 2 | 69% | No | Most frequently ranked predictive biomarker |
| | NGAL is useful for early detection of AKI | Likert | Round 1 | 85% | Yes | Widely endorsed |
| | Cystatin C improves diagnostic assessment | Ranking | Round 2 | 59% | No | Ranked second overall |
| Kidney Biopsy | Kidney biopsy is indicated for proteinuria >3 g/day | Likert | Round 1 | 52% | No | Most commonly cited biopsy trigger |
| | Preferred biopsy timing is 7–14 days after AKI onset | Likert | Round 1 | 68% | No | Early biopsy (<48h) not supported |
| | Desmopressin should be used when BUN > 100 mg/dL prior to biopsy | Likert | Round 2 | 79% | Yes | Most supported procedural recommendation |
| Contrast Use | Contrast use acceptable in CKD stage 2–3 without significant proteinuria | Likert | Round 1 | >50% | No | No formal consensus |
| | Contrast use in AKI KDIGO stage 3 | Likert | Round 1 | ~30% | No | Substantial caution among experts |
| Anemia Management in AKI | ESA and IV iron are routinely used in ICU AKI | Likert | Round 2 | 67% (rare use) | No | Majority reported infrequent use |
| Acidosis Management | Sodium bicarbonate indicated for lactic acidosis | Likert | Round 1 | 56% | No | Moderate agreement |
| | Sodium bicarbonate indicated for rhabdomyolysis | Likert | Round 1 | 52% | No | Moderate agreement |
| | THAM (tromethamine) routinely used for metabolic acidosis | Likert | Round 2 | <40% | No | Rarely used |

## Sodium bicarbonate use

Experts indicated in the survey that sodium bicarbonate was most often considered appropriate in cases of lactic acidosis (56%), and it was also commonly selected for tumor lysis syndrome (52%) and rhabdomyolysis (52%). Its role in diabetic ketoacidosis was identified much less frequently, and it was only seldom chosen for the prevention of contrast-associated AKI.

As the survey focused on identifying indications rather than therapeutic protocols, the Delphi process did not address monitoring approaches for bicarbonate administration, such as urinary pH assessment, serum markers, or titration criteria. These monitoring practices vary widely across institutions, and the consensus should therefore be interpreted as agreement on when bicarbonate may be appropriate, rather than guidance on how it should be monitored or adjusted.

### Desmopressin use

Desmopressin was mainly used before renal biopsy (24%), neurosurgery (25%), or percutaneous interventions (12%). For patients with a BUN level greater than 100 mg/dL, desmopressin was used prior to renal biopsy (40%) or neurosurgical procedures (34%). The most common dosing regimen was two doses every 12 hours, followed by two doses every 24 hours.

### Second round of Delphi survey

Statements that had not reached consensus in Round 1 were revised and presented again in Round 2. Experts rated their agreement using the same 10-point scale, and statements were presented with group-level statistics to facilitate reflection. The Round 2 survey is provided in S2 File.

### Updated ranking of AKI causes

After refining definitions, nephrotoxicity was rated as the top cause of AKI by 96% of participants. AIN followed with 40%, intratubular obstruction with 37%, and urinary obstruction with 35%.

### Biopsy indications—updated ranking

Rapidly progressive glomerulonephritis (RPGN) was the top-rated biopsy indication (72%), followed by AKI after kidney transplant (58%), non-renal solid organ transplant (45%), and bone marrow transplant (34%).

### Biomarker preferences

TIMP2-IGFBP7 was preferred by 69% of experts, followed by Cystatin C (59%) and urinary NGAL (56%).

### IV Iron and ESA use in AKI-related anemia

Intravenous (IV) iron and erythropoiesis-stimulating agents (ESAs) were infrequently used to manage anemia in ICU patients with acute kidney injury (AKI). Eighteen percent of experts reported never using IV iron or ESAs in this setting, while the majority (67%) indicated rare use. Only 2% reported routine administration.

### THAM use for acidosis

Tromethamine (THAM) was infrequently used to manage metabolic acidosis in critically ill patients. In cases of lactic acidosis, 64% of experts reported never using THAM, while 31% indicated that they used it rarely. Similar patterns were observed even in scenarios requiring high-dose inotropic support.

### Desmopressin use in ICU renal biopsy and perioperative settings

Seventy-nine percent supported using desmopressin when the BUN level was greater than 100 mg/dL. A platelet count <50,000 was the next most common indication (63%), followed by <100,000 (41%). Perioperative desmopressin use followed the same trend.

Preferred administration timing was a single pre-procedural dose (76%), with two pre-procedural doses (7%) or peri- and post-procedure dosing combinations used much less frequently.

### Summary of consensus framework

The final consensus was consolidated into a comprehensive visual framework encompassing key themes in AKI etiology, diagnostics, and ICU management (Fig 3). A detailed summary of the consensus statements, rounds achieved, and agreement percentages is provided in Table 1.

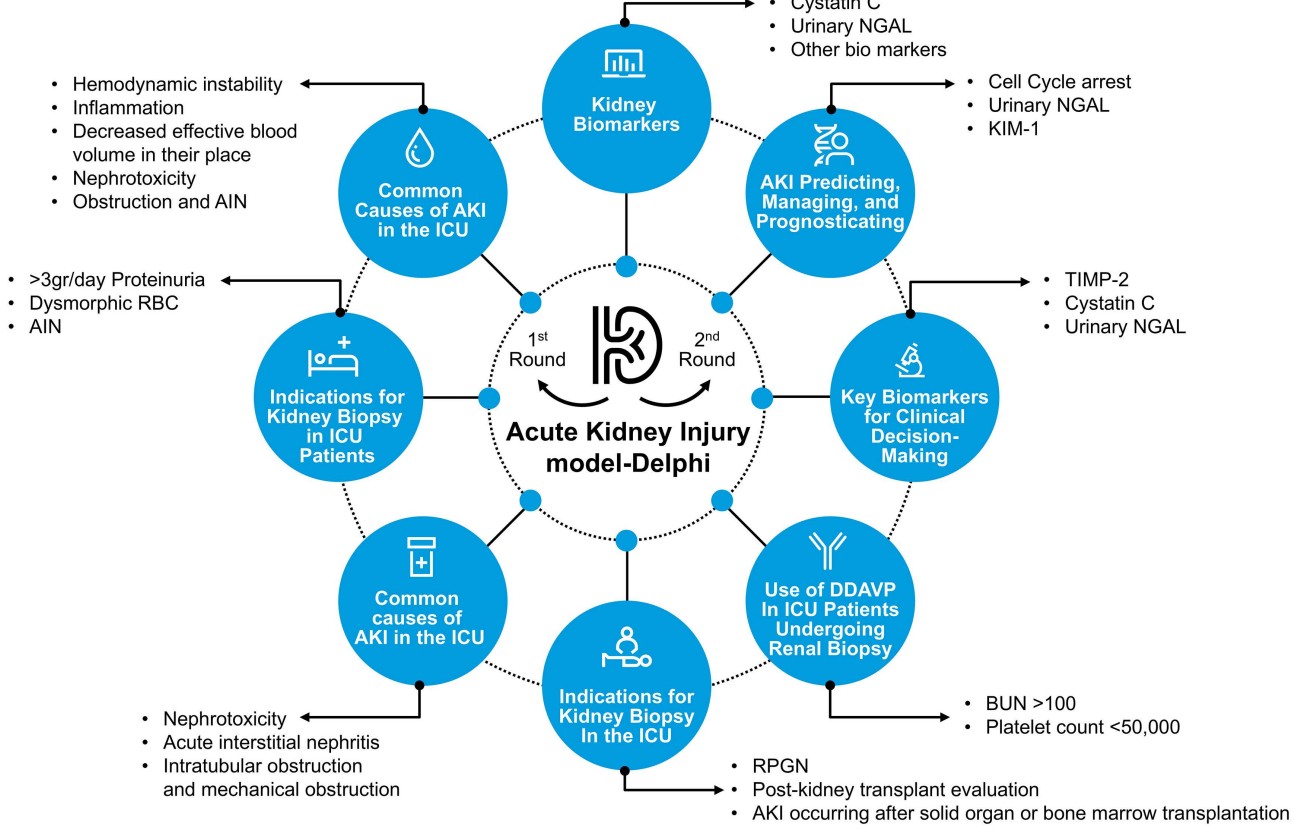

**Fig 3. Acute Kidney Injury (AKI) Delphi Model: Expert Consensus Framework.** This figure summarizes the major domains identified through the Delphi process, including etiologies, biomarker prioritization, biopsy considerations, and ICU management strategies. The biomarkers shown represent those most frequently ranked by experts for clinical decision-making, although not all reached the predefined consensus threshold.

## Directed Acyclic Graph (DAG) model

Using the finalized expert consensus statements, we constructed a simplified Directed Acyclic Graph (DAG) to represent the core causal relationships underlying acute kidney injury (AKI). Nodes in the DAG were defined based on variables that reached consensus or were consistently ranked as clinically important across Delphi rounds, including hemodynamic instability, nephrotoxicity, inflammation, cardiac dysfunction, and urinary obstruction.

Directed edges were assigned based on expert-informed causal reasoning, reflecting clinically plausible pathways through which these factors influence AKI onset and progression. Relationships were structured to maintain acyclicity, ensuring that no feedback loops were introduced and that causal flow remained unidirectional. The graph was iteratively refined by aligning its structure with the ranked consensus statements and qualitative expert feedback obtained during the Delphi process.

The resulting DAG provides a structured causal framework intended to support transparency and interpretability in subsequent AI-driven digital twin development. The final model is illustrated in Fig 4.

Consensus was defined as agreement by at least 75% of respondents, with a rating of 8 or higher on a 10-point Likert scale. Rankings reflect composite prioritization when applicable. Statements not reaching consensus but considered clinically relevant by expert commentary are also included.

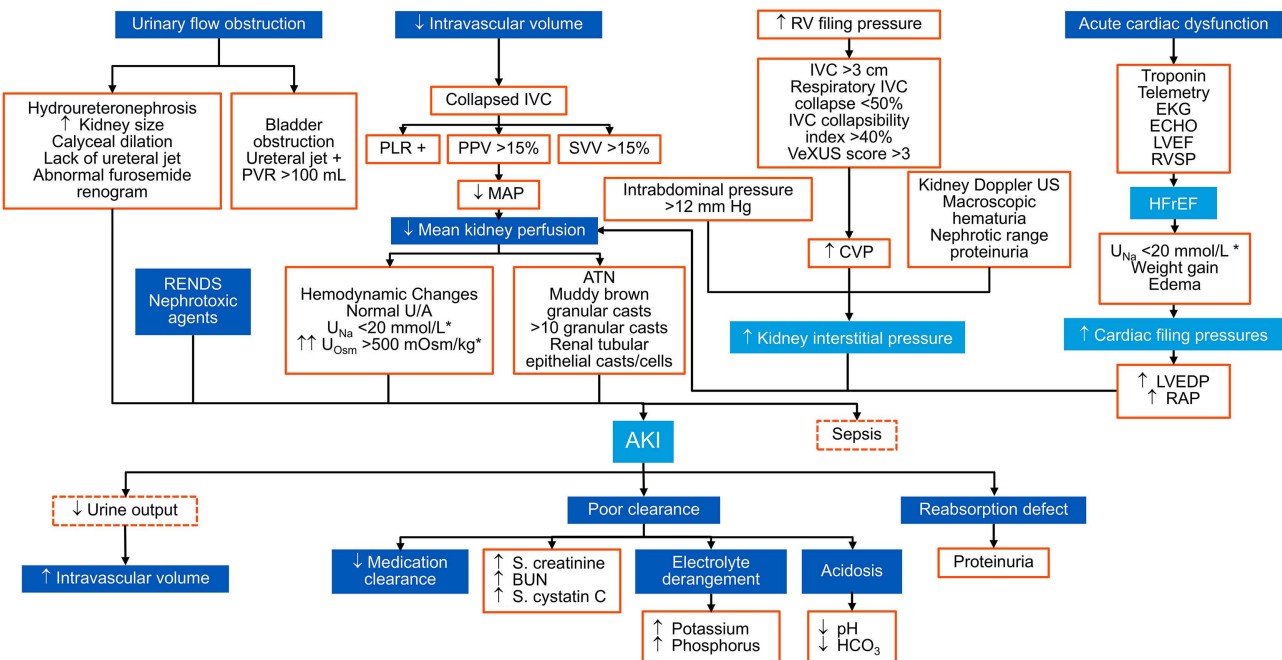

**Fig 4. AKI Causal Pathways: A Simplified Directed Acyclic Graph (DAG) Model.** This diagram illustrates the key mechanisms of AKI, highlighting the roles of hemodynamics, nephrotoxicity, cardiac dysfunction, and urinary obstruction in the progression of kidney injury. It serves as a structured framework for AI-driven modeling and clinical decision-making. The edges (or arcs) represent relationships or transitions that connect the nodes (or vertices representing entities or objects within the graph with a specific direction, i.e., they point from one node to another, implying a flow from the starting node to the target node. There are no cycles in the graph, indicating that there is no closed loop within it, which allows them to represent systems where one element flows into another in a clear, non-recursive way. DAGs in healthcare are frequently used for workflow management, data flow analysis, and Bayesian networks, in which nodes represent random variables and directed edges represent conditional dependencies between them. They are also used to assess a process's shortest path or detect cycles in the algorithms to model processes, dependencies, and data flows.

## Discussion

This international Delphi process produced a structured, clinically grounded framework for understanding acute kidney injury (AKI) in critically ill patients. Several findings align with established international recommendations, while others reveal important discrepancies between expert consensus and routine ICU practice.

The identification of hemodynamic instability and nephrotoxicity as leading contributors to AKI is consistent with epidemiological data and international guidance, including KDIGO-based frameworks and prior consensus statements [2,15,16]. In routine ICU practice, AKI frequently occurs in the setting of shock, sepsis, and systemic hypoperfusion. Although "shock" was not isolated as a discrete category in the Delphi survey, it is implicitly represented within the construct of hemodynamic instability. This may reflect the nephrology-weighted composition of the panel, which emphasizes pathophysiological mechanisms over syndromic ICU classifications. From a critical care perspective, AKI is often conceptualized within broader hemodynamic syndromes such as septic or cardiogenic shock, which may not have been fully captured as independent entities in this consensus.

The biomarker findings warrant particular consideration. A strong majority of experts endorsed incorporating biomarkers beyond serum creatinine and urine output, consistent with ongoing research highlighting the potential diagnostic and prognostic value of markers such as neutrophil gelatinase–associated lipocalin (NGAL), cystatin C, and cell cycle arrest markers (TIMP-2·IGFBP7) [6,9,10,14,17].

 

However, routine ICU practice remains more conservative. In many centers, AKI management continues to rely primarily on creatinine trends, urine output, electrolyte panels, and clinical assessment. Advanced biomarkers are not universally available and are inconsistently integrated into bedside decision-making. The Delphi responses may therefore reflect academic and forward-looking perspectives rather than uniform real-world implementation. This gap underscores the translational challenge between biomarker validation studies and widespread clinical adoption.

Panel composition represents an important limitation. Although the group included intensivists and dual-trained clinicians, only 24% were intensivists without nephrology specialization. This imbalance may have influenced prioritization toward renal-specific mechanisms rather than broader hemodynamic and resuscitation strategies that dominate general ICU workflows. For example, contrast-enhanced imaging is frequently performed in critically ill patients when diagnostic urgency outweighs theoretical nephrotoxicity risks. Contemporary ICU practice often involves individualized risk–benefit assessment rather than strict avoidance of iodinated contrast, particularly in life-threatening scenarios. Similarly, routine monitoring commonly relies on serum creatinine, urea, ionograms, and urinary indices rather than advanced biomarker panels. These differences highlight how subspecialty composition may shape consensus outcomes.

The discussion surrounding desmopressin also merits contextualization. Consensus supported its use in patients with blood urea nitrogen levels above 100 mg/dL prior to kidney biopsy, yet broader ICU practice varies considerably. Evidence guiding desmopressin administration in critically ill patients remains limited, and its use is often individualized rather than protocolized. The relatively strong endorsement observed in this Delphi may reflect nephrology procedural norms more than general ICU bleeding management practices.

Geographic distribution further influences interpretation. The predominance of United States–based experts may partially explain certain practice patterns, including enthusiasm for biomarker integration and procedural approaches. Access to diagnostic tools, institutional resources, and medico-legal frameworks differ internationally and may shape clinical decisions in ways not fully captured by this panel.

Despite these limitations, the structured expert-derived framework remains valuable. The purpose of this Delphi process was not to replace prospective trials or observational datasets, but to formalize clinically reasoned causal architecture that can inform interpretable modeling. Directed Acyclic Graphs (DAGs) were used to translate consensus relationships into an explicit structure suitable for subsequent digital twin development [17–20]. Rather than functioning as a predictive model itself, the DAG provides a transparent scaffold clarifying assumed causal pathways among exposures, mediators, and outcomes. This approach complements data-driven modeling by embedding domain expertise into algorithm design.

Importantly, expert consensus represents structured clinical reasoning rather than empirical validation. Future research must integrate prospective data, external validation cohorts, and outcome-based evaluation to determine whether such consensus-informed frameworks improve predictive accuracy, clinical utility, and patient-centered outcomes.

## Conclusion

This study brought together an international panel of experts to develop a unified and clinically grounded framework for understanding acute kidney injury in the critical care setting. Through a structured Delphi process, we achieved consensus on the key contributors, diagnostic considerations, and management strategies that shape AKI care in the ICU. These insights were then translated into a simplified causal model that reflects how experienced clinicians interpret and respond to this complex condition.

By organizing expert knowledge into a format that supports real-time clinical reasoning, this work helps bridge the gap between clinical expertise and the early development of AI-driven digital twin models. The resulting framework offers more than a summary of current practice. It provides a foundation for designing interpretable, context-aware tools that can incorporate patient-specific complexity and help promote consistency in care.

At the same time, expert consensus alone is insufficient to guide clinical decision-making. High-quality prospective studies remain essential to confirm and refine therapeutic approaches. Ongoing and upcoming clinical trials, including investigations such as BICAR-2, will continue to play a critical role in shaping evidence-based AKI management. This framework is therefore intended to complement, rather than replace, the insights gained from rigorous clinical research.

Taken together, the consensus structure presented here is an initial platform that can be strengthened as new data emerge. It highlights the value of collaboration across disciplines. It underscores the importance of integrating expert judgment with prospective clinical evidence as we work toward the next generation of personalized, data-driven AKI care.

## Supporting information

**S1 File. Delphi Round 1 questionnaire.**
(PDF)

**S2 File. Delphi Round 2 questionnaire.**
(PDF)

**S3 File. Minimal dataset underlying the findings, including description, de-identified item-level responses, summary statistics, and consensus determinations for Delphi Rounds 1 and 2.**
(XLSX)

## Author contributions

**Conceptualization:** Jacob Ninan, Kianoush Kashani.

**Data curation:** Mehdi Kashani, Jacob Ninan, Kianoush Kashani.

**Formal analysis:** Mehdi Kashani, Kianoush Kashani.

**Funding acquisition:** Mehdi Kashani.

**Investigation:** Mehdi Kashani, Wisit Cheungpasitporn, Kianoush Kashani.

**Methodology:** Mehdi Kashani, Jacob Ninan, Kianoush Kashani.

**Project administration:** Jacob Ninan, Kianoush Kashani.

**Resources:** Kianoush Kashani.

**Software:** Amos Lal, Kianoush Kashani.

**Supervision:** Wisit Cheungpasitporn, Ognjen Gajic, Kianoush Kashani.

**Validation:** Ognjen Gajic, Kianoush Kashani.

**Visualization:** Mehdi Kashani, Jacob Ninan, Lifang Wei, Wisit Cheungpasitporn, Amos Lal, Kianoush Kashani.

**Writing – original draft:** Mehdi Kashani, Jacob Ninan, Lifang Wei.

**Writing – review & editing:** Mehdi Kashani, Jacob Ninan, Wisit Cheungpasitporn, Amos Lal, Ognjen Gajic, Kianoush Kashani.

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
