## [Decision Letter · Decision Letter 0]

11 Feb 2026

International Delphi Consensus on Acute Kidney Injury: Foundations for AI-Driven Digital Twin Development in Critical Care Nephrology

PLOS One

Dear Dr. Kashani,

Thank you for submitting your manuscript to PLOS ONE. After careful consideration, we feel that it has merit but does not fully meet PLOS ONE’s publication criteria as it currently stands. Therefore, we invite you to submit a revised version of the manuscript that addresses the points raised during the review process.

We look forward to receiving your revised manuscript.

Kind regards,

Chiara Lazzeri

Academic Editor

PLOS One

Journal Requirements:

2. We note that Figure 2 in your submission contains map images which may be copyrighted. All PLOS content is published under the Creative Commons Attribution License (CC BY 4.0), which means that the manuscript, images, and Supporting Information files will be freely available online, and any third party is permitted to access, download, copy, distribute, and use these materials in any way, even commercially, with proper attribution. For these reasons, we cannot publish previously copyrighted maps or satellite images created using proprietary data, such as Google software (Google Maps, Street View, and Earth). For more information, see our copyright guidelines: http://journals.plos.org/plosone/s/licenses-and-copyright.

1. You may seek permission from the original copyright holder of Figure 2 to publish the content specifically under the CC BY 4.0 license.

3. We note you have included a table to which you do not refer in the text of your manuscript. Please ensure that you refer to Table 1 in your text; if accepted, production will need this reference to link the reader to the table.

4. We note that Supporting Information 1 and 2 in your submission contain copyrighted images (logos). All PLOS content is published under the Creative Commons Attribution License (CC BY 4.0), which means that the manuscript, images, and Supporting Information files will be freely available online, and any third party is permitted to access, download, copy, distribute, and use these materials in any way, even commercially, with proper attribution. For more information, see our copyright guidelines: http://journals.plos.org/plosone/s/licenses-and-copyright.

1. You may seek permission from the original copyright holder of Supporting Information 1 and 2 to publish the content specifically under the CC BY 4.0 license.

Reviewers' comments:

Reviewer's Responses to Questions

**Comments to the Author**

1. Is the manuscript technically sound, and do the data support the conclusions?

Reviewer #1: Yes

Reviewer #2: Yes

2. Has the statistical analysis been performed appropriately and rigorously?

Reviewer #1: N/A

Reviewer #2: Yes

3. Have the authors made all data underlying the findings in their manuscript fully available?

Reviewer #1: Yes

Reviewer #2: Yes

4. Is the manuscript presented in an intelligible fashion and written in standard English?

Reviewer #1: Yes

Reviewer #2: Yes

Reviewer #1: This article by Kashani et. al is a DELPHI Consensus on AKI in critical care with A DAG in order to build a digital twin model for simulation.

This concept is appealing, in an area of research that needs to be developed to improve simulation and teaching.

The paper is well written and pleasing to read. However it lacks some detailed results on the DELPHI and a real discussion on the results itself.

Minor revisions:

- Line 74: all while reducing patient risk by � missing word ?

- Line 110-111: The fact that you have reached consensus after two rounds belong to the results section. I would only describe your method in the method section. A maximum of three rounds planned to reach 75% agreement among experts, etc… In the result section, you can mention that agreement was achieved after 2 rounds. Same for the abstract.

- Line 119: add references to support one’s statement

- Line 136 : how many participants had AI related clinical initiatives ?

- Line 193 to 206: please define abbreviations of biomarkers when first used in the text

- Line 217: Define RBCs and AIN abbreviations

- Line 222: Contrast use in CKD and AKI patients is not mentionned again in the Table 1 nor the second round of delphi. Could it be possible to give the results for AKI KDIGO III ? Which is the most frequent situation encountered in the ICU.

- Line 266: please define abbreviation when first used in the text (THAM)

- Line 280-285: give more details on the DAG and on its construction/logic.

- Figure 3: It is hard to know what the first and second round with arrows refers to in the figure. Does that mean that consensus was obtained in the first round for the items on the left and items on the right for the second round ?

- Figure 3: How did you define key biomarkers for clinical decision making ? was it a specific item of the DELPHI ? It does not appear in the Table 1 nor in the main text.

- Table 1: I would add more details of the DELPHI results in the table, for each item and question.

Major revisions:

The paper is interesting and serves a bigger purpose. However, the discussion lacks feedback on the results of the DELPHI itself.

1- Discussion on the results: do the result of the Delphi match international recommendations ? does it match routine clinical ICU practice ? That would be particularly interesting for the biomarkers (I would be surprised if ICU routinely use NGAL Cystatin C TIMP2 IGFBP7 …)

2- Discuss the panel of experts, only 24% are intensivists, without a dual specialty in Nephrology. It really shows in the results of the Delphi. Most AKI in the regular ICU is the result of shock, which is not detailed in the paper. From a critical care perspective, the contrast use is widely used and outweighs the cons of contrast in AKI, we usually use simple biomarkers (creatinine, urea, ionogram, urinary ionogram). It is a severe limitation. Another limitation of the panel is the majority of americans experts (which is mentionned in the discussion). It could explain certain practices that seem unusual for a non-american intensivist.

3- Discuss the use of Desmopressin

4- Some part of the discussion on the relevance of a Delphi and a DAG for a digital twin is redundant, and could be shortened.

Reviewer #2: Kashani et al provide their manuscript titled 'International Delphi Consensus on Acute Kidney Injury: Foundations for AI-Driven Digital Twin Development in Critical Care Nephrology'. With this Delphi the authors were able to identify key items to be used for an artificial intelligence (AI) model in patients with acute kidney injury. I think this manuscript could be of added value, although it is not possible to evaluate the AI model itself.

I have some minor comments.

Introduction

Line 74: sentence is suddenly stopped, please revise.

Methods

- Line 113: could you provide a reference to substantiate this?

- What is the number of experts that was approached?

**Do you want your identity to be public for this peer review?** For information about this choice, including consent withdrawal, please see our Privacy Policy

Reviewer #1: No

Reviewer #2: No

---

## [Author Response · Author response to Decision Letter 1]

23 Feb 2026

Manuscript ID: PONE-D-25-64532

Title: International Delphi Consensus on Acute Kidney Injury: Foundations for AI-Driven Digital Twin Development in Critical Care Nephrology

Dear Editor and Reviewers,

We sincerely thank the Academic Editor and both reviewers for their thoughtful and constructive evaluation of our manuscript. We greatly appreciate the time and expertise invested in reviewing our work.

Before addressing the reviewer comments, we confirm that we have ensured full compliance with PLOS ONE’s formatting and licensing requirements. Specifically:

• Documentation confirming licensing compliance for Figure 2 has been uploaded with this revision.

• All institutional and platform logos have been removed from the Supporting Information files to comply with CC BY 4.0 policies.

• Table 1 is now appropriately cited in the manuscript text.

• The manuscript has been formatted according to PLOS ONE guidelines.

• All major and minor revisions are clearly highlighted in red in the revised manuscript.

Response to the Academic Editor

In response to the Editor’s request to strengthen the discussion of the Delphi results, we have substantially revised and expanded the Discussion section. Specifically, we have:

• Provided deeper contextualization of the Delphi findings relative to international recommendations and routine ICU practices.

• Clarified the translational gap between biomarker endorsement and real-world ICU implementation.

• Expanded discussion of contrast use in AKI, desmopressin practices, and the influence of panel composition (including subspecialty distribution and geographic representation).

• Streamlined redundant sections related to the Delphi–DAG–digital twin framework to improve clarity and focus.

All supplementary information regarding the figures are uploaded.

This study is based on a Delphi process which is survey of experts anonymously. We did not include any patient information and all data from the surveys are already available in the manuscript.

Response to Reviewer #1

We thank Reviewer #1 for the positive evaluation of the scientific rigor and conceptual importance of our work.

All minor comments have been addressed, including 1) correction of the incomplete sentence in the Introduction section, 2) clarification of the Delphi methodology by separating planned design from achieved results, 3) addition of methodological references supporting the consensus threshold, 4) definition of all abbreviations at first use, 5) clarification of biomarker terminology, 6) expanded reporting of KDIGO stage III findings, 7) definition of THAM, and 8) detailed description of the DAG construction process.

Figure 3 has been revised for clarity, and the figure legend has been updated to explicitly explain the framework and avoid confusion regarding round-specific arrows and biomarker categorization.

Table 1 has been expanded to provide detailed round-by-round results, including agreement percentages and consensus status.

In response to the major comments, we substantially revised the Discussion section to critically analyze the Delphi findings, compare them with international recommendations and routine ICU practice, discuss limitations related to panel composition and geographic predominance, explicitly address desmopressin use, and reduce redundancy in the discussion of the digital twin conceptual framework.

Response to Reviewer #2

We thank Reviewer #2 for the positive assessment of the manuscript’s scientific validity.

We corrected the incomplete sentence in the Introduction section, added appropriate methodological references, and clarified the total number of experts invited and participating in the Delphi process.

We believe these revisions have substantially strengthened the manuscript’s clarity, methodological transparency, and clinical contextualization. We are grateful to the reviewers for their insightful comments, which have significantly improved the rigor and presentation of our work.

Sincerely,

Kianoush B. Kashani, MD, MS

Corresponding Author

---

## [Editor Report · Decision Letter 1]

2 Mar 2026

International Delphi Consensus on Acute Kidney Injury: Foundations for AI-Driven Digital Twin Development in Critical Care Nephrology

PONE-D-25-64532R1

Dear Dr. Kashani,

We’re pleased to inform you that your manuscript has been judged scientifically suitable for publication and will be formally accepted for publication once it meets all outstanding technical requirements.

Kind regards,

Chiara Lazzeri

Academic Editor

PLOS One
---

## [Editor Report · Acceptance letter]

PONE-D-25-64532R1

PLOS One

Dear Dr. Kashani,

I'm pleased to inform you that your manuscript has been deemed suitable for publication in PLOS One. Congratulations! Your manuscript is now being handed over to our production team.

Kind regards,

on behalf of

Dr. Chiara Lazzeri

Academic Editor

PLOS One